# Single-molecule analysis of steroid receptor and cofactor action in living cells

Ville Paakinaho[1,*], Diego M. Presman[1,*], David A. Ball[1], Thomas A. Johnson[1], R. Louis Schiltz[1], Peter Levitt[1], Davide Mazza[2], Tatsuya Morisaki[1,†], Tatiana S. Karpova[1] & Gordon L. Hager[1]

Population-based assays have been employed extensively to investigate the interactions of transcription factors (TFs) with chromatin and are often interpreted in terms of static and sequential binding. However, fluorescence microscopy techniques reveal a more dynamic binding behaviour of TFs in live cells. Here we analyse the strengths and limitations of *in vivo* single-molecule tracking and performed a comprehensive analysis on the intranuclear dwell times of four steroid receptors and a number of known cofactors. While the absolute residence times estimates can depend on imaging acquisition parameters due to sampling bias, our results indicate that only a small proportion of factors are specifically bound to chromatin at any given time. Interestingly, the glucocorticoid receptor and its cofactors affect each other's dwell times in an asymmetric manner. Overall, our data indicate transient rather than stable TF-cofactors chromatin interactions at response elements at the single-molecule level.

[1] Laboratory of Receptor Biology and Gene Expression, National Cancer Institute, National Institutes of Health, Building 41, 41 Library Drive, Bethesda, Maryland 20892, USA. [2] Istituto Scientifico Ospedale San Raffaele, Centro di Imaging Sperimentale e Università Vita-Salute San Raffaele, 20132 Milano, Italy. * These authors contributed equally to this work. † Present address: Department of Biochemistry and Molecular Biology, Colorado State University, Fort Collins, Colorado 80523, USA. Correspondence and requests for materials should be addressed to G.L.H. (email: hagerg@exchange.nih.gov).

For decades, transcription factors (TFs) have been thought to operate by binding their genomic targets stably to form well-defined macromolecular complexes that remain intact and biologically functional for minutes or hours at a time[1–3]. This 'assembly-function-dissociation model' emerged from *in vitro* reconstitution and *in vivo* population assays such as chromatin immunoprecipitation (ChIP). Under this paradigm, the recruitment of multiple molecular partners progressively stabilizes the structure and facilitates the recruitment of other factors in a static and well-ordered manner[4]. However, this view was challenged when fluorescent tagged proteins enabled the visualization of TF dynamics in live cells[5]. Perturbation assays such as fluorescence recovery after photobleaching (FRAP) or correlation techniques like fluorescent correlation spectroscopy (FCS) unequivocally show a more complex picture[2,6]. Almost all factors that have been studied by live-cell microscopy exhibit dwell times at chromatin on the order of seconds[7]. Hence, an alternate model wherein dynamics plays a central role is taking centre stage. According to this new view, the key to efficient recruitment of the transcription machinery to its target site relies on two fundamental dynamic properties of TFs: their ability to rapidly diffuse through the nucleus and their propensity to bind transiently to chromatin[8].

With recent advances in fluorescence imaging, it has become possible to track individual TF molecules in single live cells[6,9–16]. This provides a methodology for elucidating the search pattern and efficiency of TFs in finding and binding to their target sites[11,12]. Steroid receptors are a class of ligand-inducible TFs that respond to environmental stimuli and mediate the expression of genes involved in metabolic, developmental and inflammatory pathways[17]. Their hormone-dependent nature makes these proteins ideal for studying TF dynamics. In this work, we compared the dynamic behaviour of steroid receptors at the single-molecule level using single-molecule tracking (SMT). In particular, we focused on the relationship between the glucocorticoid receptor (GR) and several of its cofactors.

In this study, we show that only a small percentage of TF molecules ($\sim$5–10%) are specifically bound to chromatin at any given time. We also show that, given the multi-exponential nature of the distribution of residence times, the measured kinetic parameters for TF binding can depend on the image acquisition parameters. Therefore, the dwell time for a given factor must be evaluated over a range of conditions to develop a rigorous understanding of its dynamic behaviour. Finally, we report an asymmetric modulation between GR and its known cofactors GRIP1, BRG1 and AP-1 at the single-molecule level. Collectively, our single-molecule studies affirm the general model where many transcription factors are highly dynamic during their chromatin-binding activity.

## Results

**Interpretation of residence times and bound populations.** In SMT, individual protein molecules are imaged in time-lapse and their spatial-temporal trajectories are recorded in real time[18] (Fig. 1a). Such single molecules produce a diffraction-limited spot in the image, which can be localized with a precision below the diffraction limit of $\sim$200 nm (ref. 2). Initial tracking of individual GR molecules was performed under highly inclined and laminated optical sheet (HILO) illumination[19] by sub-optimal transient transfection of HaloTag-fused GR (HaloTag-GR)[10] labelled with the Janelia Fluor 549 (JF$_{549}$) HaloTag ligand[20].

Subsequently, the cells were treated with the natural ligand corticosterone (Cort) prior to imaging. Using a custom-made tracking software (see Methods), individual localizations of single molecules are classified as belonging to a bound track segment (that is, moving less than a defined threshold $r_{max}$ for more than

$N_{min}$ consecutive frames) or unbound (see Methods). From the bound population, we could then compute the duration of each binding event: we observed a continuum of residence times exponentially distributed (Fig. 1b), ranging from 0.2 to 47.4 s. A single-exponential model is insufficient to fit the data, while a two-component model fits the data with high precision (Fig. 1c). Interestingly, a three-fit component provides no improvement over the two-fit component (Fig. 1c), suggesting the presence of only two distinct populations of bound molecules. This has previously been interpreted as the TF binding to non-specific sites in chromatin (fast component), and to specific response elements (slow component)[12]. The average residence time that is extracted from the exponential decay model (Fig. 1g) should not be mistaken with the average number that one can obtain from a Gaussian-distributed population. In a symmetric normal distribution, the mean represents both the most likely value (that is, the mode) and the central tendency of the distribution (that is, the median). However, when the population is exponentially distributed, the mean does not represent the most likely value.

To further understand the nature of the fast versus slow binding events, we mutated the DNA-binding domain of GR. The mutation C440G severely compromises the first zinc-finger within GR's DNA-binding domain (DBD), therefore abolishing specific DNA-binding activity *in vitro*[21]. Interestingly, HaloTag-GRC440G-Cort molecules still present a slow fraction component but with a threefold decreased bound fraction (0.8%) compared to wild type (2.6%) (Fig. 1h; Supplementary Fig. 1a,k). The effect is very similar to inactivating the wild-type receptor by washing-out the ligand (Cort-wash, Fig. 1j; Supplementary Fig. 1e,g). Since GR is fully dimeric *in vivo*[22] and recently shown to form tetramers after DNA binding[23], it is possible that the endogenous GR aids the mutant by forming heterodimers before chromatin binding (Fig. 1k). We therefore utilized the monomeric GR mutant (GRmon)[22] in combination with the DBD mutation (GRmonC440G). The HaloTag-GRmonC440G-Cort manifests no slow fraction component (Fig. 1i,l, Supplementary Fig. 1b,l), clearly supporting the model that the long-lived molecules indeed represent specific binding to DNA.

Multiple studies have shown that different GR ligands can influence the mobility and binding of GR to chromatin[24–26]. Consistently, the synthetic agonist dexamethasone (Dex) promotes a longer residence time[10] and increased bound fraction compared to Cort (Fig. 1d; Supplementary Fig. 1f,h). Surprisingly, both HaloTag-GRC440G and HaloTag-GRmonC440G still present a small long-lived binding component when activated by Dex (Fig. 1e,f; Supplementary Fig. 1c,d,i,j). These results suggest that some slow component binding to chromatin may be achieved in a DBD-independent manner. Binding modes such as protein–protein 'tethering' are frequently discussed as alternate mechanisms for site-specific factor localization[27]. Collectively (Supplementary Fig. 1m), our data strongly support the model that long-lived binding represents specific interaction with chromatin (that is, specific recognition sequences).

**Sampling bias in single-molecule tracking.** To test whether the HaloTag-GR chimera is functional, we first knocked-out the endogenous GR in 3617 cells (KOGR, Fig. 2a). Subsequently, we stably integrated HaloTag-GR into the KOGR cell line (KOGR + HaloTag-GR, Fig. 2a). These cells present a Dex-dependent mRNA increase of known GR target genes (Fig. 2c), indicating that the HaloTag-GR is transcriptionally active. Interestingly, SMT analysis shows that, although there is a significant increase in residence time ($P < 0.001$), the bound

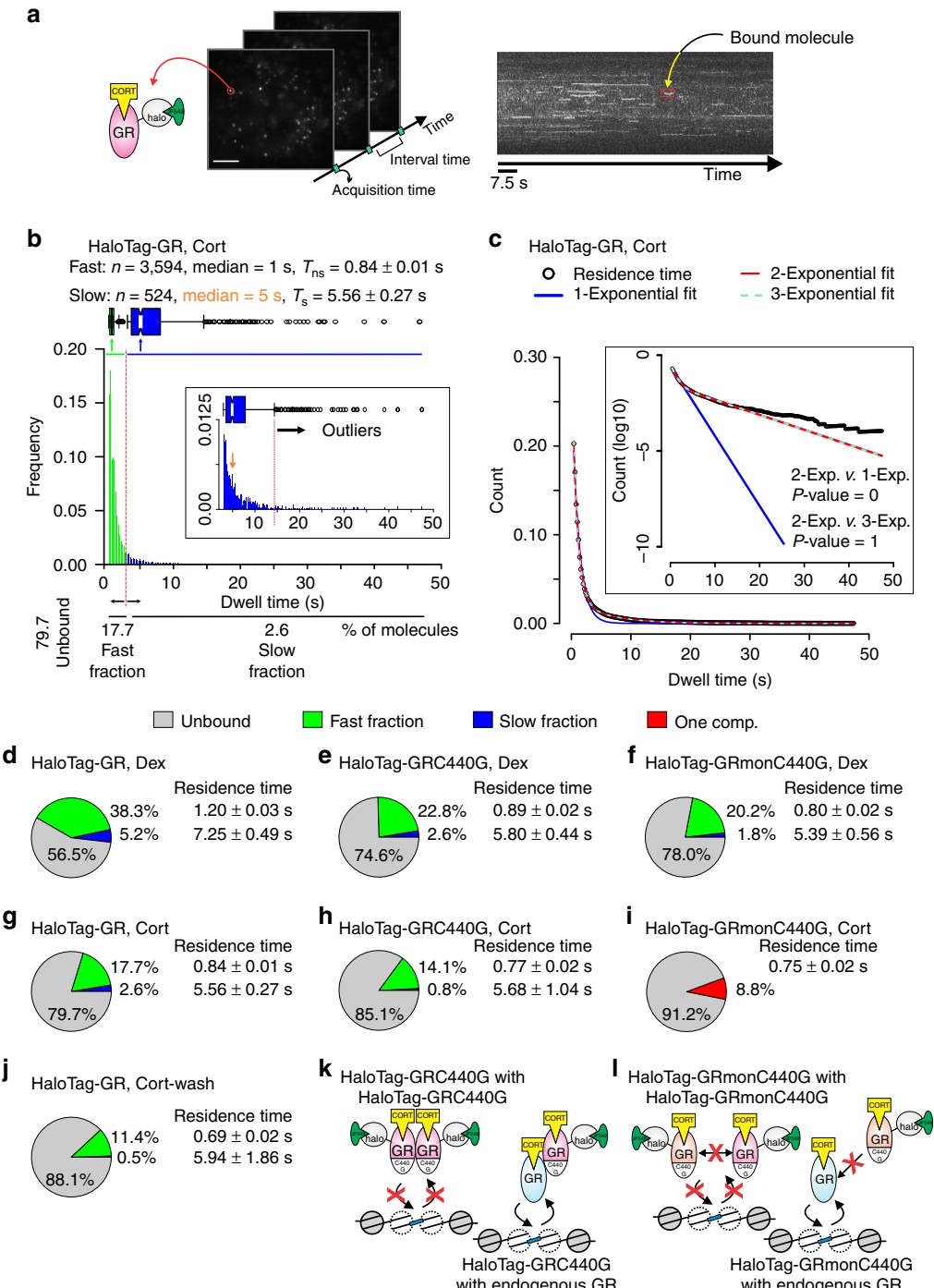

**Figure 1 | Single-molecule tracking (SMT) of GR molecules.** (**a**) The SMT technique visualizes individual molecules as bright diffraction-limited spots and tracks their movement or lack thereof over time. HaloTag-GR (cartoon) labelled with Janelia Fluor 549 (JF$_{549}$) can be visualized as such diffraction-limited spots under HILO microscopy. Cort, corticosterone. Scale bar, 5 μm. A stack of images is taken from a single live cell with a fixed acquisition time and a specific interval time. If molecules remain stationary, the time-projection stack will reveal a continuous signal that represents a bound molecule (red box). (**b**) Distribution of residence times from individual GR( + Cort) stationary tracks, either in a histogram or in a Box-plot. A continuum of bi-exponentially distributed bound molecules is typically observed, based on the fitting of the survival distribution. The fast short-lived ($T_{ns}$, non-specific) and slow long-lived ($T_s$, specific) fractions are colour-coded (green and blue, respectively). Inset shows only the $T_s$ population (orange arrow, median). The number (n) of tracks obtained, and the median dwell time in $T_{ns}$ and $T_s$ fraction is shown above the histogram. (**c**) Single molecules of GR( + Cort) data represented as collected tracks (black circles) in a survival distribution plot, fitted to a single- (blue line), double- (red line) or three-exponential (dashed light blue line) decay model. Inset view with y axis plotted as a log10. F-test determines the statistical significance of the fit between different decay models. (**d–j**) Pie-charts represent percentage of molecules unbound (grey), bound at the fast, short-lived fraction (green), and bound at the slow, long-lived fraction (blue) of HaloTag-GR under different conditions as indicated. In the case of HaloTag-GRmonC440G( + Cort) (**i**), a single-exponential (one component, red) was sufficient to explain the data. The average residence time of fast, short-lived and slow, long-lived fraction is presented next to their representative fractions. (**k,l**) The cartoons illustrate the likely 'interference' between HaloTag-GR and the endogenous GR molecules. Exposure time 10 ms; interval time 200 ms.

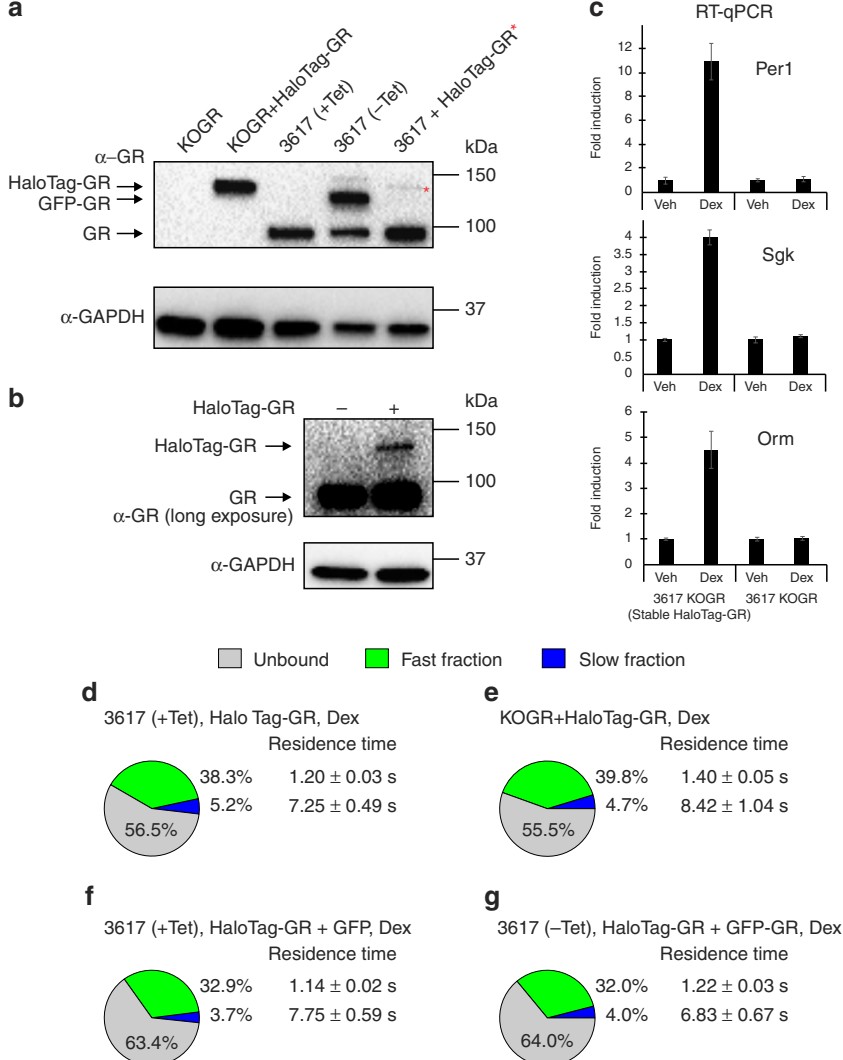

**Figure 2 | Single-molecule behaviour of transiently expressed of HaloTag-GR reflects well the behaviour of stably integrated HaloTag-GR.**
(**a**) Immunoblotting against GR showing the different protein levels in 3617 cells knockout of endogenous GR (KOGR), stably integrated HaloTag-GR cells (KOGR + HaloTag-GR), 3617 cells treated with ( + Tet) or without (-Tet) tetracycline to suppress or express stably integrated GFP-GR, and 3617 cells transiently transfected with HaloTag-GR. The band representing HaloTag-GR in transiently transfected conditions is marked with a red asterisk. Immunoblotting with GAPDH antibody was used as a loading control. (**b**) Immunoblotting with GR antibody with a longer exposure time reveals the lower levels of transiently transfected HaloTag-GR compared to the endogenous protein. (**c**) mRNA expression of GR target genes, Period 1 (*Per1*), serum and glucocorticoid-regulated kinase (*Sgk*), and orosomucoid (*Orm*) after 1 h Dex treatment in 3617 knockout of endogenous GR (3617 KOGR) and 3617 KOGR with stably integrated HaloTag-GR (3617 KOGR Stable HaloTag-GR) cells. Bar graph represent mean fold induction ± s.d. Data represent at least two biological replicates. (**d–g**) The bound fractions and the average residence time for transiently transfected HaloTag-GR in 3617 cells (**d**), stably integrated HaloTag-GR in KOGR cells (**e**), transiently transfected HaloTag-GR + GFP in 3617 cells (**f**), and transiently transfected HaloTag-GR in 3617 cells grown without Tet to induce expression of stably integrated GFP-GR (**g**). Pie charts presented as in Fig. 1. Exposure time 10 ms; interval time 200 ms.

fraction is very similar ($P = 0.41$) between stably integrated and transiently transfected HaloTag-GR (Fig. 2d,e; Supplementary Fig. 1f,h); despite the fact that the average expression levels in the transient transfections are lower than in the stable integrated cell line (Fig. 2b). Furthermore, if we artificially increase GR levels by inducing the GFP-GR transgene in 3617 cells (Fig. 2a), we do not observe any drastic change in the slow bound fraction compared to the GFP alone control ($P = 0.67$) (Fig. 2f,g). Overall, results indicate that the transient transfection of HaloTag-GR accurately reflects the conditions of a stably integrated transgene.

Live-cell binding studies using FCS, FRAP and SMT techniques have yielded widely divergent estimates of the chromatin-bound fraction of TFs and their residence times[6]. While FCS and FRAP are indirect methods and rely on kinetic models to obtain these

parameters, SMT is more straightforward as bound molecules are directly visualized[2]. Ideally, to achieve the maximum amount of dynamic information, one should acquire images as fast as technically possible. However, this will lead to a faster bleaching of individual fluorophores and consequently to an underestimation of the residence time. SMT experiments similar to those described above, where 10 ms exposures are interleaved with 200 ms of 'dark time' could hamper the detection of highly stable single molecules, considering that we could on an average follow individual molecules for 190 frames before photobleaching (Supplementary Table 1). Hence, a careful balance between temporal resolution and dynamic range must be obtained.

Therefore, to detect whether highly stable single molecules exist, we tracked HaloTag-GR in Dex-treated cells under a wide

range of interval acquisition times (Fig. 3a). Acquisition of the data from 30 ms to 1.5 s interval times consistently gives three populations of molecules: (i) unbound, (ii) fast bound (green boxes) and (iii) slow bound (blue boxes). We observed that longer imaging intervals produce longer average TF residence times. These averages are widely spread from a mean of ∼2.6 s to up to ∼37 s for the slow component (Supplementary Table 1; Supplementary Data 1). We interpret these results as follows:

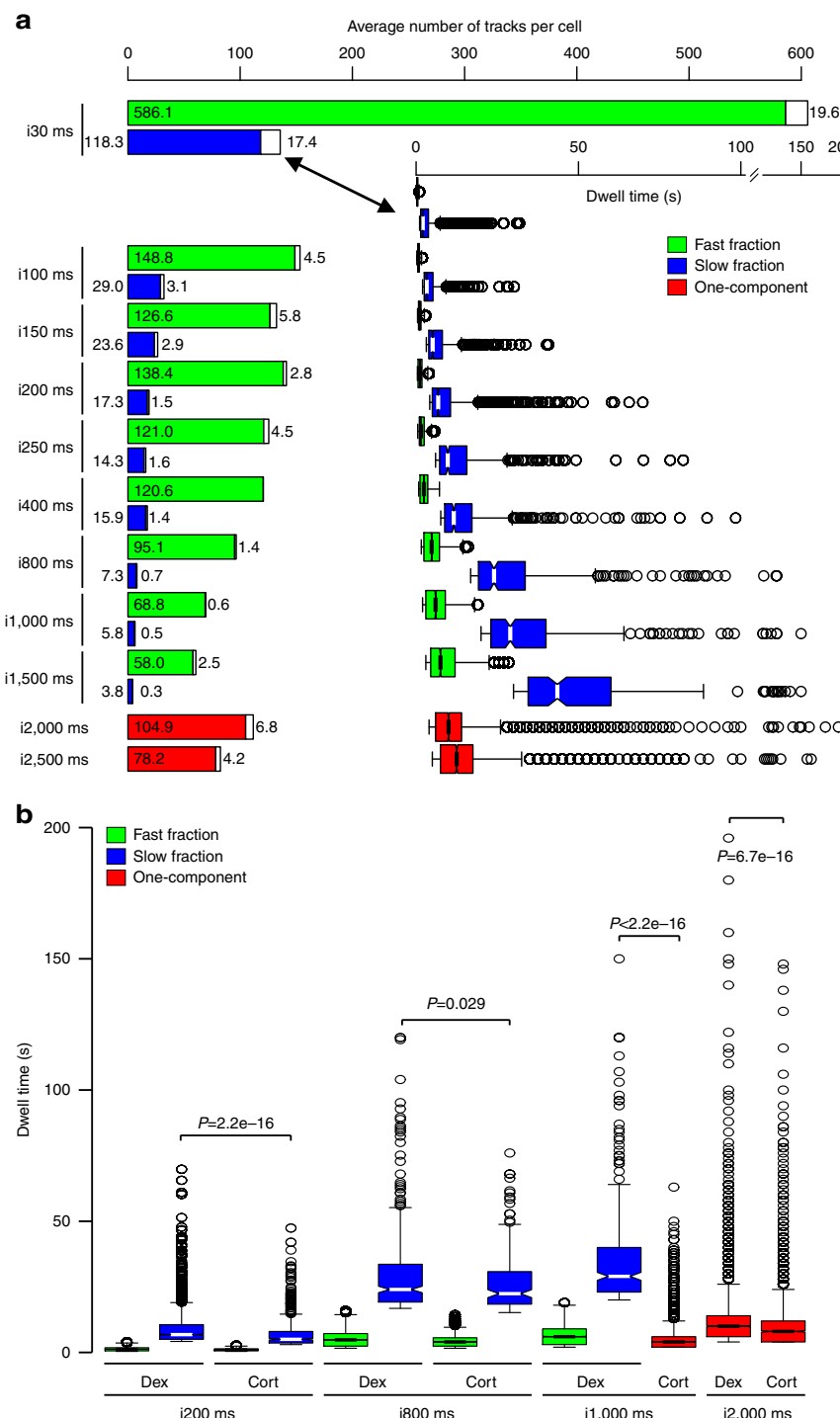

**Figure 3 | HaloTag-GR single-molecule tracking data with different interval times.** (**a**) Acquisition parameter effects. SMT of HaloTag-GR labelled with JF$_{549}$ in Dex-treated cells as a function of different interval acquisition times. The figure shows the average number of tracks per cell on each fraction and the Box-plot data of residence times under the indicated interval times. White bars represent the average number of outliers within each population. For each condition, the data have been corrected for photobleaching. (**b**) Comparison of dwell time distribution of HaloTag-GR treated with Dex or Cort under different interval times as indicated. Box-plot represents the distribution of dwell times for all molecules in the fast short-lived (green boxes) and slow long-lived (blue boxes) fraction for GR. GR in Cort-treated cells with 1,000 and 2,000 ms interval time and GR in Dex-treated cells with 2,000 ms interval time are presented as one-component fraction (red boxes). Statistical outliers are shown as black circles. *P* values represent a Two-sample Kolmogorov–Smirnov test defined by the brackets. Exposure time 10 ms; interval time as indicated.

when using fast acquisition parameters, bleaching becomes the predominant factor in SMT and the most stable binding events are no longer recorded; hence, residence time is underestimated. On the contrary, when longer imaging intervals are used, faster transiently bound molecules cannot be accurately tracked; or worse, they can be mistaken for a longer track if two independent events appear and disappear during the 'dark' time. Hence, there is an overestimation of the residence times. When imaging intervals are too sparse to track the vast majority of fluorescent molecules, the slow fraction is defined primarily by the outliers observed at faster frame rates (compare 1,500–2,000 ms in Fig. 3a, right graph). In agreement, longer interval times produce vastly fewer tracks recorded per cell (Fig. 3a, left graph). In other words, for each image acquisition rate, different populations of molecules are being sampled. Remarkably, when intervals are increased to 2 and 2.5 s, the transiently bound GR is no longer detected and a single-component exponential decay model is sufficient to describe the system (Fig. 3a, red boxes and Supplementary Movie 1). If GR presented very long and stable binding events, a second population should have been detected under these conditions (Supplementary Table 1), as has been recently observed for the telomerase protein at 1 s interval[28]. Taken together, these results suggest it is very unlikely that long, highly stable molecules exist in the GR bound population. Importantly, differences between treatments are still observed under different acquisition conditions, as exemplified by comparing Dex and Cort effect across different interval times (Fig. 3b and Supplementary Data 1). Furthermore, the single residence time obtained at the 2 s interval is similar to the one found at the 200–250 ms intervals for the slow fraction (Fig. 3a and Supplementary Data 1), indicating that the latter range is appropriate for capturing the majority of the dynamics of this particular system. Hence, all comparisons below were performed under the 200 ms acquisition time.

**Activation of steroid receptors alter their binding dynamics**. To study the changes in binding dynamics of steroid receptors before and after activation, we tracked HaloTag-fused GR, oestrogen (ER), progesterone (PR), and androgen (AR) receptors in the 3617 cell line in both untreated and cognate hormone-treated conditions. A two-exponential decay model was required to characterize each receptor in all conditions (Supplementary Fig. 2). For each receptor (Fig. 4a), the vast majority of unliganded molecules are diffusing and remain unbound (grey colour), which reflects the inactive status of the TF. Each receptor displayed a varying degree of short-lived (fast fraction, green colour) and long-lived (slow fraction, blue colour) binding events. However, in the case of untreated HaloTag-GR and -PR, the long-lived events are insignificant, comparable to that of HaloTag alone (Supplementary Fig. 3a). In contrast, both Halo-Tag-ER and -AR showed a higher fraction of both short-lived (>20%) and long-lived (>2%) binding events in the untreated condition, compared to that of HaloTag alone. Furthermore, the fraction of long-lived binding events of HaloTag-ER and -AR are significantly higher ($P < 0.05$) than that of HaloTag-GR and -PR. It is tempting to speculate this could reflect TF activity of the unliganded forms of ER and AR, as previously reported elsewhere[29,30].

Activation of all steroid receptors increased the fast and slow bound fractions, while the proportion of unbound molecules was reduced (Fig. 4a,b; Supplementary Fig. 3b–i). Significant increases in long-lived residence times after activation are observed with HaloTag-ER, HaloTag-GR and HaloTag-PR (Fig. 4a,c). Interestingly, HaloTag-AR does not show an increase in the long-lived residence time after hormone treatment. However, the

proportion of long-lived binding events of AR after activation is significantly increased (Fig. 4a,b). A similar rise in binding events with varying degrees is also observed with the other steroid receptors. Comparison of activated steroid receptors shows that HaloTag-ER differs from the others by having a higher bound fraction and residence time (Fig. 4b,c). The contrast of steroid receptors in untreated versus activated conditions in live cells is consistent with the known biological properties of these TFs; hormonal activation leads to chromatin binding of ER, GR, PR and AR (Fig. 4d), and subsequent regulation of transcription[31–34]. The increase in both residence time and long-lived binding events after hormone treatment suggests that these more stable, yet dynamic binding events likely represent functional chromatin-binding.

**GRIP1-binding dynamics are influenced by GR activation**. We next analysed how known GR partners would respond to receptor activation. To this end, we tracked HaloTag-fused GR-interacting protein 1 (GRIP1), and SNAP-tag fused BRG1, a major ATPase subunit of the SWI/SNF chromatin remodelling complex. GRIP1 is a well-known coactivator of GR[35,36], while BRG1 has a major role in regulating accessibility at GR-bound enhancers[37,38]. Intranuclear single molecules of HaloTag-GRIP1 and SNAP-tag-BRG1 were tracked in both untreated and Dex-treated conditions (Supplementary Fig. 4a–f). Similar to the receptors, a two-exponential decay model was required to characterize each cofactor (Supplementary Fig. 5). In untreated conditions, HaloTag-GRIP1 shows a relatively high bound fraction and residence time (Fig. 5a). As expected, GRIP1 is likely associated with multiple other TFs prior to GR activation. However, endogenous GR activation with Dex still results in a significant increase in both the fraction and residence time of long-lived HaloTag-GRIP1 molecules (Fig. 5a,b,e,f). Thus, GR induces a larger frequency and more stable recruitment of GRIP1 to chromatin. In comparison, SNAP-tag-BRG1 long-lived binding events remain unchanged after Dex-treatment (Fig. 5a).

To confirm that the increase in GRIP1's residence time and bound fraction is due to the direct interaction of GR with GRIP1, we tracked HaloTag-fused GRIP1 interaction-defective mutant[39] (GRIP1mut; Supplementary Fig. 4c,d). In untreated conditions, GRIP1 and GRIP1mut show similar bound fraction and residence time with no significant changes (Fig. 5a,b,e,f). In contrast to wild-type HaloTag-GRIP1, both residence time and bound fraction of GRIP1mut remains unchanged after Dex treatment (Fig. 5e,f; Supplementary Fig. 4g). This indicates that the increase in residence time and long-lived binding events of GRIP1 is due to the interaction of GR with GRIP1. Next, we wondered whether depletion of GRIP1 would affect the residence time or bound fraction of GR. We used siRNA to knock-down GRIP1 levels (Supplementary Fig. 4h), and subsequently tracked HaloTag-GR in the presence of control siRNA (siSCR) or GRIP1 siRNA (siGRIP1). Results surprisingly show an asymmetric relationship, since neither GR's residence time nor the specific bound fraction is affected by GRIP1's absence (Fig. 5c,d,g,h; Supplementary Fig. 4i). Moreover, GR transactivation activity was not altered under these conditions (Supplementary Fig. 4j).

**AP-1-binding dynamics are not influenced by GR activation**. The GR-dependent changes in GRIP1 action suggest that GR can alter the binding dynamics of important co-regulators. A major regulator of GR action in 3617 cells is AP-1, which modulates GR binding at 40% of its binding sites[32]. In addition, GR is capable of regulating AP-1 binding but to a lesser extent. To investigate GR and AP-1 interplay at the single-molecule level,

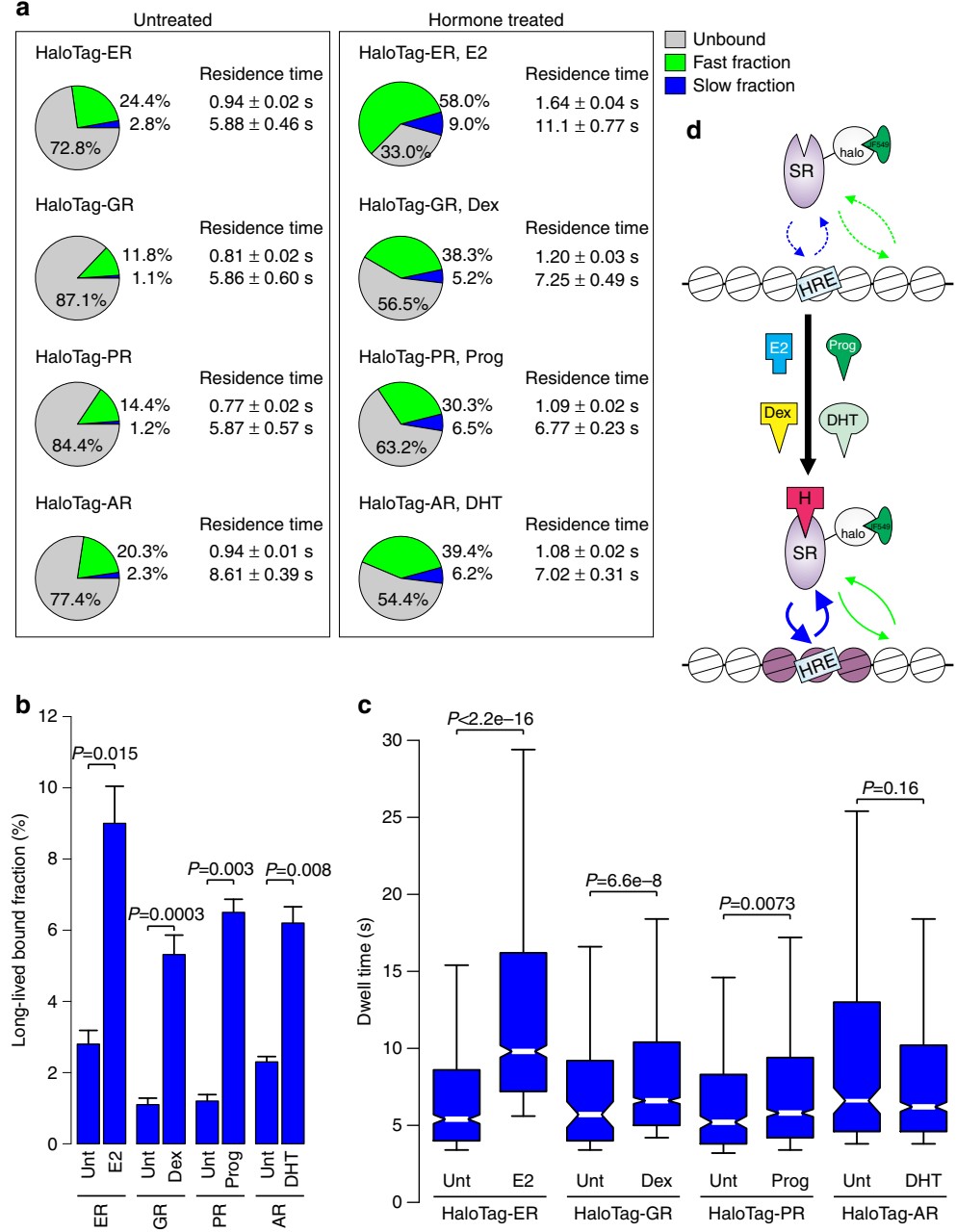

**Figure 4 | Single-molecule analysis indicates an increase in residence time and bound fraction of steroid receptors after hormonal activation.**
(**a**) The bound fractions and the average residence time for HaloTag-ER, -GR, -PR and -AR in untreated cells (left panel), and in cognate hormone-treated cells (17β-estradiol (E2) for ER, Dex for GR, Progesterone (Prog) for PR, and dihydrotestosterone (DHT) for AR) (right panel). Pie charts presented as in Fig. 1. (**b**) Bar chart represents the long-lived fraction for ER in untreated and E2-treated cells, GR in untreated and Dex-treated cells, PR in untreated and Prog-treated cells, and AR in untreated and DHT-treated cells. $P$ values from a Student's $t$-test defined by the brackets. Bar graph represents the mean long-lived bound fraction ± s.d. Data collected from at least two independent microscopy sessions. The number of cells and tracks analysed per condition are described in Supplementary Data 1. (**c**) Box-plot represents the distribution of dwell times for all molecules in the slow long-lived fraction for ER in untreated and E2-treated cells, GR in untreated and Dex-treated cells, PR in untreated and Prog-treated cells, and AR in untreated and DHT-treated cells. $P$ values represent a Two-sample Kolmogorov–Smirnov test defined by the brackets. (**d**) Cartoon illustrates activation of the receptor leads to a decrease in unbound and increase in bound receptors. H, steroid hormone; HRE, hormone response element; SR, steroid receptor. Exposure time 10 ms; interval time 200 ms.

we tracked the major subunits of AP-1, c-JUN and c-FOS[40,41] fused with HaloTag in untreated and Dex-treated conditions (Supplementary Fig. 6a–d). A two-exponential decay model was required to characterize c-JUN and c-FOS (Supplementary Fig. 6f–i). Prior to GR activation, both HaloTag-c-FOS and -c-JUN show relatively slow residence times in the long-lived slow fraction (Fig. 6a). Interestingly, HaloTag-c-JUN presents the slowest residence time of all factors tested, and has the lowest percentage of unbound molecules (<40%). Dex-treatment produces no noticeable effect on the residence time (Fig. 6a,c; Supplementary Fig. 6k) or the long-lived binding events (Fig. 6a,b) of HaloTag-c-FOS and -c-JUN.

The dominant negative a-FOS, which contains an acidic extension in its N terminus, effectively inhibits the DNA binding of c-JUN while having no effect on the heterodimerization of c-JUN-a-FOS[42]. To study the effect of a-FOS on AP-1 action at the single-molecule level, we tracked HaloTag-fused a-FOS (Supplementary Fig. 6e,j). HaloTag-a-FOS shows significant decrease in the proportion of molecules in the slow fraction compared to that of c-FOS (Fig. 6a,b), and the residence time is

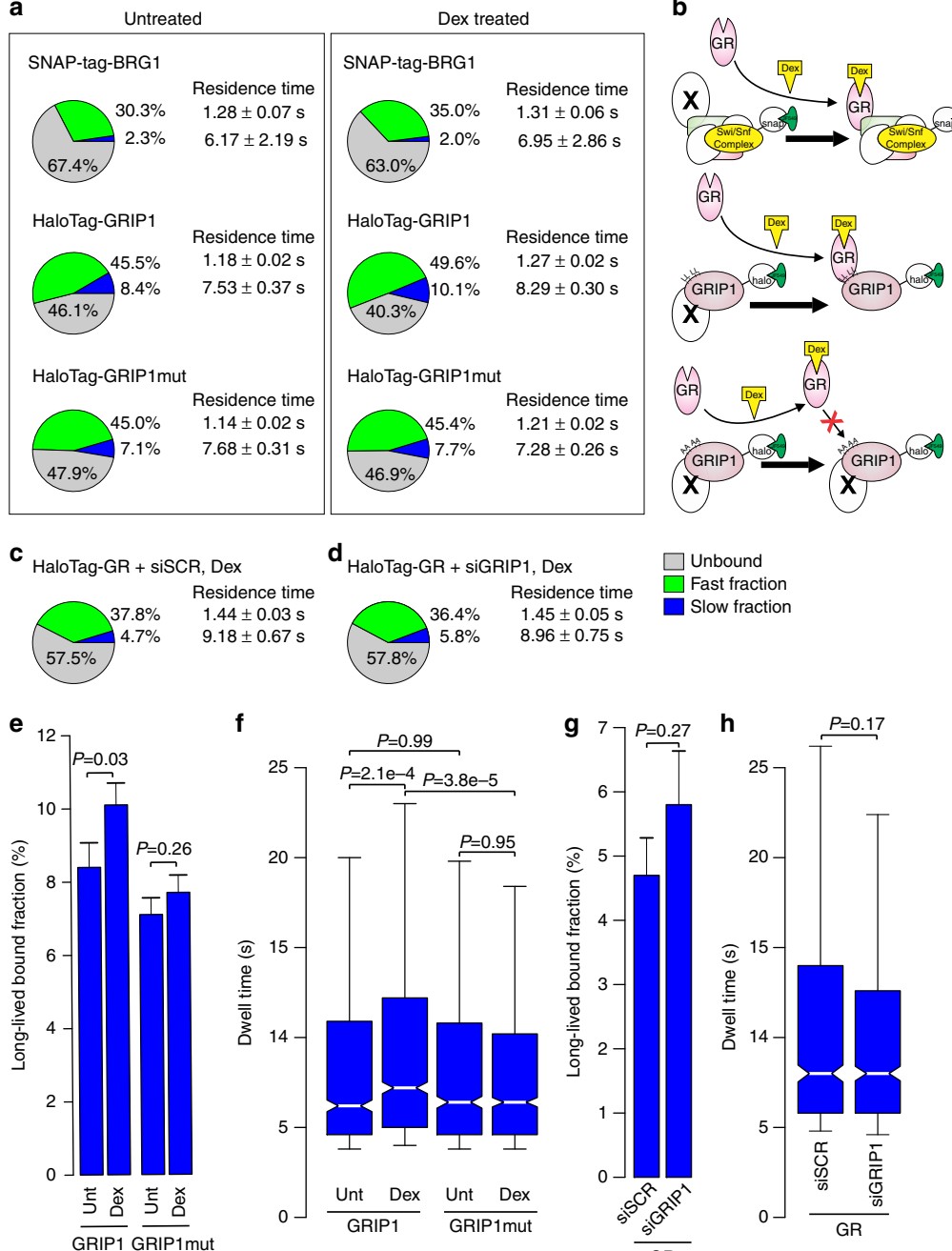

**Figure 5 | Asymmetric modulation between GR and cofactors at the single-molecule level.** (**a**) The bound fractions and the average residence time for HaloTag-GRIP1, -GRIP1mutant (GRIP1mut) and SNAP-tag-BRG1 in untreated and in Dex-treated cells. Pie charts presented as in Fig. 1. (**b**) Cartoons of the respective pie charts illustrate the interaction of GR with GRIP1 and BRG1 after Dex treatment, and the inhibition of interaction with GRIP1mut. (**c,d**) The bound fractions and the average residence time for HaloTag-GR in Dex-treated cells exposed to control siRNA (siSCR) (**c**), or GRIP1 siRNA (siGRIP1) (**d**). (**e**) Bar chart represents the long-lived fraction for GRIP1, and GRIP1mut in untreated and Dex-treated cells. *P* values represent a Student's *t*-test defined by the brackets. Bar graph represents the mean long-lived bound fraction ± s.d. Data collected from at least two independent microscopy sessions. The number of cells and tracks analysed per condition are described in Supplementary Data 1. (**f**) Box-plot represents the distribution of dwell times for all molecules in the long-lived fraction for GRIP1, and GRIP1mut in untreated and Dex-treated cells. *P* value represents a Two-sample Kolmogorov–Smirnov test defined by the brackets. (**g**) Bar chart represents the long-lived fraction for GR in siSCR- and siGRIP1-treated cells. *P* values represent a Student's *t*-test defined by the brackets. (**h**) Box-plot represents the distribution of dwell times for all molecules in the long-lived fraction for GR in siSCR- and siGRIP1-treated cells. *P* value represents a Two-sample Kolmogorov–Smirnov test defined by the brackets. Exposure time 10 ms; interval time 200 ms.

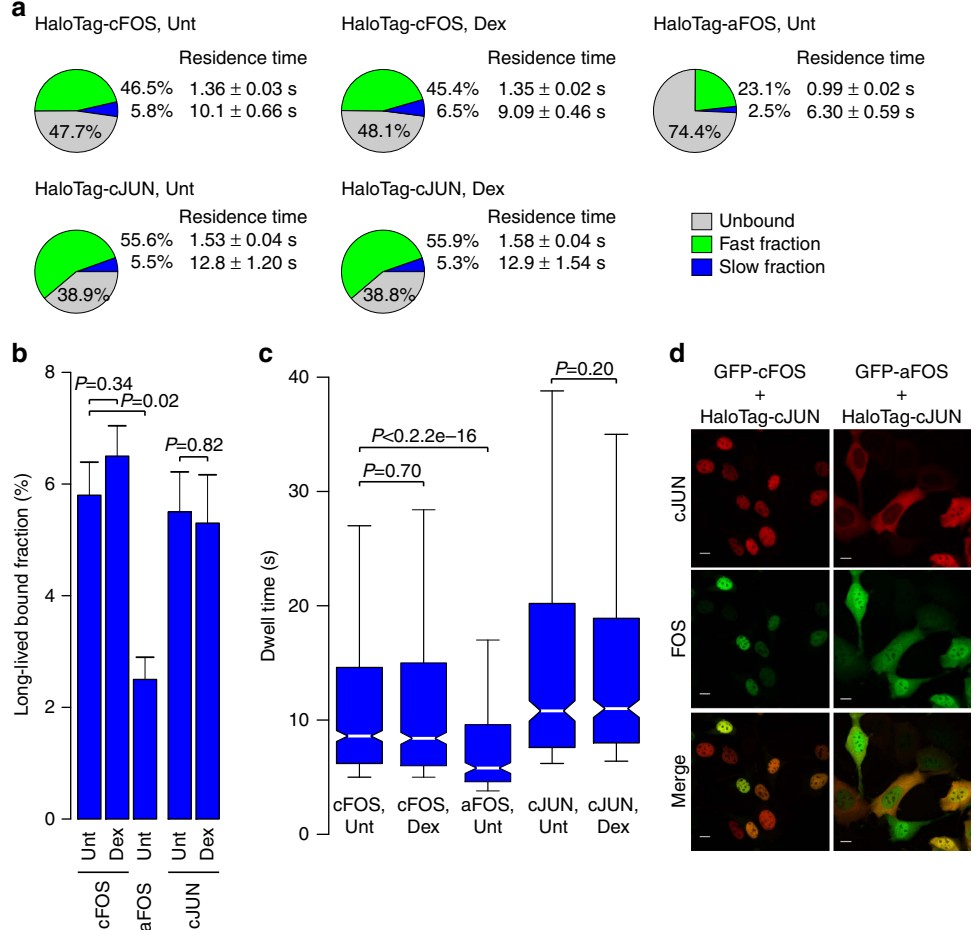

**Figure 6 | The residence time and bound fraction of c-JUN and c-FOS are Dex-independent.** (**a**) The bound fractions and the average residence time for HaloTag-c-FOS, -a-FOS and c-JUN in untreated cells, and HaloTag-c-FOS, and -c-JUN in Dex-treated cells. Pie charts presented as in Fig. 1. (**b**) Bar chart represents the long-lived fraction for c-FOS, a-FOS and c-JUN in untreated and c-FOS, and c-JUN in Dex-treated cells. P values represent a Student's t-test defined by the brackets. Bar graph represents the mean long-lived bound fraction ± s.d. Data collected from at least two independent microscopy sessions. The number of cells and tracks analysed per condition are described in Supplementary Data 1. (**c**) Box-plot represents the distribution of dwell times for all molecules in the long-lived fraction for c-FOS, a-FOS and c-JUN in untreated, and c-FOS and c-JUN in Dex-treated cells. P values represent a Two-sample Kolmogorov–Smirnov test defined by the brackets. Exposure time 10 ms; interval time 200 ms. (**d**) Confocal microscopy images of cells expressing HaloTag-c-JUN with GFP-c-FOS (left panels) or GFP-a-FOS (right panels). Top panels represent c-JUN (red), middle panels FOS (green) and lower panels merge of c-JUN and FOS. Scale bar, 5 μm.

significantly decreased (Fig. 6a,c). Interestingly, we were unable to measure the intranuclear dynamics of HaloTag-c-JUN in the presence of a-FOS, as most of the HaloTag-c-JUN molecules were exported out of the nucleus (Fig. 6d). In the presence of GFP-c-FOS, HaloTag-c-JUN co-localizes with c-FOS in the nucleus. However, in the presence of GFP-a-FOS, HaloTag-c-JUN is found mainly in the cytoplasm, while a-FOS is in both compartments. Dimerization of c-FOS and c-JUN is known to inhibit the nuclear exit of AP-1 (ref. 43), indicating that in addition to inhibiting the DNA binding of c-JUN, a-FOS influences the nuclear export of c-JUN.

**AP-1 affects the single-molecule binding dynamics of GR.** ChIP-seq analyses have shown that AP-1 modulates GR action by being an initiator of GR binding[32]. To investigate how AP-1 influences GR binding dynamics, we tracked HaloTag-GR in the presence of GFP-a-FOS (Supplementary Fig. 7a,b). Again, a two-exponential decay model was required to characterize GR in each condition (Supplementary Fig. 7c,d). Slight but significant

changes were observed for the HaloTag-GR long-lived residence time in the presence of GFP and GFP-c-FOS (Figs 2f and 7a,d; Supplementary Fig. 7e). Notably, the long-lived bound fraction does not change between the two control conditions (Figs 2f and 7a,c). However, in the presence of GFP-a-FOS, GR shows a clear reduction in the long-lived bound fraction, and a significant decreased in residence time (Fig. 7). In fact, the proportion of single molecules in the long-lived bound fraction decreases to almost the same level as HaloTag-GR in untreated conditions (Fig. 4a). Moreover, GR dynamics in the presence of a-FOS are comparable to DBD mutants of GR (compare Fig. 1e,f with Fig. 7b,e and Supplementary Fig. 7f). This demonstrates that proper action of AP-1 is crucial for the dynamic binding of GR. These SMT findings are markedly consistent with the genomic data showing an initiating role for AP-1 in GR binding[32].

**Discussion**
Two mutually exclusive views have been proposed regarding TF dynamics on chromatin: the genomic view, based on *in vitro*

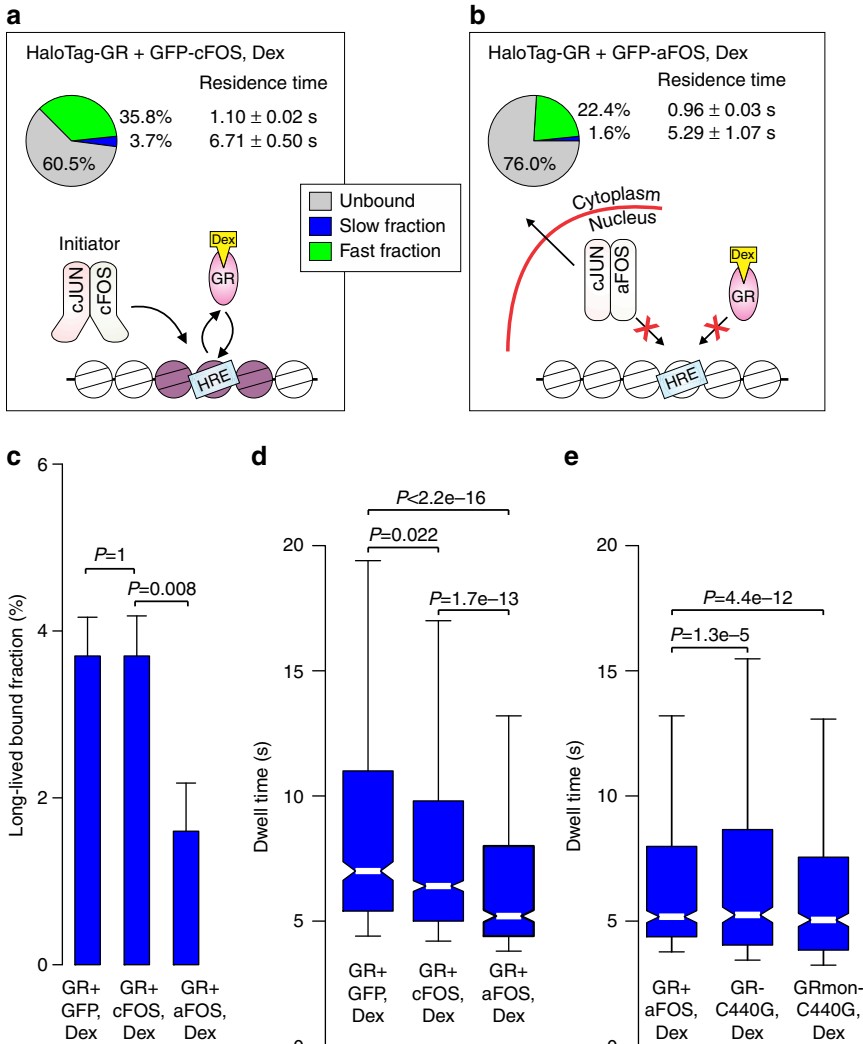

**Figure 7 | Disruption of AP-1 reduces GR residence time and bound fraction to a similar level as GR DNA-binding mutant.** (**a,b**) The bound fractions and the average residence time for HaloTag-GR in the presence of GFP-c-FOS (**a**), and GFP-a-FOS (**b**) in Dex-treated cells. GFP was used to visualize the FOS-positive cells. Pie charts presented as in Fig. 1. Cartoons below the pie charts illustrate the importance of AP-1 in regulating GR chromatin binding. (**c**) Bar chart represents the long-lived fraction for GR + GFP, GR + GFP-c-FOS and GR + a-FOS in Dex-treated cells. *P* values represent a Student's *t*-test defined by the brackets. Bar graph represents the mean long-lived bound fraction ± s.d. Data collected from at least two independent microscopy sessions. The number of cells and tracks analysed per condition are described in Supplementary Data 1. (**d**) Box-plot represents the distribution of dwell times for all molecules in the long-lived fraction for GR + GFP, GR + GFP-c-FOS and GR + a-FOS in Dex-treated cells. *P* values represent a Two-sample Kolmogorov–Smirnov test defined by the brackets. (**e**) Box-plot represents the distribution of dwell times for all molecules in the long-lived fraction for GR + a-FOS, GRC440G and GRmonC440G in Dex-treated cells. *P* values represent a Two-sample Kolmogorov–Smirnov test defined by the brackets. Exposure time 10 ms; interval time 200 ms.

biochemistry with naked DNA and *in vivo* ChIP-seq approaches, postulates that TFs are sequentially and stably (minutes–hours) recruited to DNA. Conversely, the live cell microscopy view, grounded on fluorescence techniques such as FCS, FRAP and more recently SMT, holds that TFs are transiently bound (milliseconds–seconds) to the template. For a comprehensive understanding of TF action, the genomic and live cell microscopy perspectives must eventually be resolved within a single comprehensive model. Genomic studies indicate that hormonal activation results in binding of receptors to chromatin[31–34]. The SMT data reported here are consistent in several aspects with the genomic studies. Activation of ER, GR, PR and AR with cognate hormones leads to an increase in the long-lived fraction and to a decrease in the fraction of the unbound molecules. Moreover, binding is more stable in activated conditions compared to the

untreated state. However, SMT results diverge markedly from population-based assays in the underlying dynamics. It must be emphasized that the long binding times inferred from these approaches represent an interpretation of static experiments, not from direct measurements. For the steroid receptor bound fraction, we observed a continuum of residence times exponentially distributed and ranging from the milliseconds range to dozens of seconds. No evidence of very long, stable binding was observed. Although there are statistically classified outliers that present individual binding times in the 30–40 s range, these represent ∼0.25% of the entire population. In yeast, it has been previously demonstrated that this low number of events cannot be solely responsible for transcriptional activation[44].

Several lines of evidence now point to the 'slow fraction' measured by single-molecule experiments as the molecules

interacting with authentic regulatory sites. Introduction of mutations that abolish DNA binding *in vitro* drastically affects the long-lived binding events of several TFs *in vivo*[6,12,14]. Moreover, these slow binding events are only observed in the presence of RNA polymerase II foci[10]. Here we found that for Cort-activated GR, all long-binding events are abolished if GR's DBD is no longer functional, and the possibility of heterodimerization with endogenous molecules is eliminated (Cort-GRmonC440G). This argues in favour of the specific binding nature of this sub-population. Furthermore, disruption of one of the master accessibility factors for GR binding (that is, AP-1)[32] severely affects GR's slow stops, mimicking DNA-binding defective conditions. This is also consistent with the model wherein AP-1 regulates GR binding by dynamic assisted loading[45]. It is intriguing that without a functional DBD or heterodimerization with the endogenous GR, a very small fraction of long-lived binding events is still observed for the Dex-GRmonC440G. This population may in fact represent GR tethering with other TFs to some regions in the genome, as suggested by ChIP-exo data[46]. Alternatively, this could also reflect non-chromatin binding to 'assembly factories' or foci, as previously speculated[47,48].

We also demonstrate that the estimates of average residence time can depend on the imaging acquisition parameters, when multi-exponential distributions of residence times are observed. Hence, determination of absolute average residence times should be taken with caution. Previous reports have attempted to deal with this issue with no clear resolution of the problem[9,49,50]. Here we have taken an empirical approach to deal with this potential issue: by collecting data with different intervals between images, we have identified parameters that allow the collection of information both on the 'fast fraction' and on the 'slow fraction' of binding events, and also to appreciate differences in the residence time distribution between different experimental conditions. Given the technical difficulties of providing and exact number for the wide distributions of residence times observed, together with the expectation that different response elements would have different residence times[10] it is possible that this single parameter may oversimplify complexity in the interaction of TFs with transcription sites. Nevertheless, at least relative comparisons of average residence times between different proteins, under the same acquisition conditions, are still valid. For example, c-FOS and c-JUN clearly show longer residence times compared to GR (compare Figs 2d and 6a), while FoxA1 is slightly faster than ER[16].

Binding characteristics of BRG1 resemble to those of a TF, consistent with the concept that complexes are formed between Swi/Snf proteins and various TFs within the nucleus[51]. We failed to observe any drastic changes in the behaviour of BRG1 at the single-molecule level when GR is activated. As this chromatin remodeler interacts with many TFs, it is likely that any effect of GR constitutes a small fraction of all the interactions of BRG1 and gets averaged out. In striking contrast, GRIP1 showed an increase in bound fraction and residence time of the long-lived component when GR is activated, and this effect depends on direct GR–GRIP1 interactions. This suggests that GR significantly reorganizes the binding landscape of GRIP1, as previously shown by genomic approaches[36]. Nevertheless, it is noteworthy that a relatively large fraction of GRIP1 molecules are pre-bound independently of GR, suggesting that GRIP1 has multiple other TF partners. As the GR uses many p160 members as cofactors[52], redundancy would explain why GRIP1 depletion did not have any effect on GR activity, residence time or bound fraction. Alternatively, since GRIP1 has also been characterized as a corepressor[17], effects on transactivation may not be expected.

Our single-molecule imaging data support the concept that many TFs as well as different co-regulators are highly dynamic during their chromatin-binding activity[6,9–12,14,16]. In all cases, only a small proportion of molecules seem functionally bound at any given time. The dynamic nature of TF binding suggests that rather than simultaneously occupying the same stretch of DNA, these proteins exchange rapidly at response elements.

In conclusion, TF dwell times at authentic response elements are quite brief for the increasing number of factors that have been investigated. These studies must now be extended to the direct analysis of dynamics at known regulatory sites, and the corresponding linkage to real-time transcriptional output for promoter(s) dependent on those sites. Although this is an ambitious goal, it is likely technically possible with the rapidly evolving methodology available in live cell microscopy.

## Methods

**Cell lines and cell culture.** Dexamethasone (Dex), corticosterone (Cort), Progesterone (Prog), 17β-estradiol (E2) and Dihydrotestosterone (DHT) were purchased from Sigma-Aldrich (St Louis, MO, USA). All cell lines were routinely cultured in DMEM high-glucose supplemented with 10% fetal bovine serum (FBS) (Life Technologies, Grand Island, NY, USA) and 2 mM L-glutamine (Life Technologies).

3617 mouse mammary adenocarcinoma cells[5], unless otherwise indicated, were grown in the presence of 5 µg ml$^{-1}$ tetracycline (Sigma-Aldrich) to prevent expression of a stably integrated GFP-GR[22]. To stably express their integrated GFP-GR, 3617 cells were grown without tetracycline for 24 h. Prior to hormone treatments, cells were seeded into two-well Lab-Tek chamber slides (Thermo Fisher, Waltham, MA, USA), transiently transfected and incubated for at least 18 h in DMEM medium containing 10% charcoal-stripped FBS (Life Technologies) and 2 mM L-glutamine. Knock-out of GR in 3617 cells (KOGR cells) was achieved by CRISPR-Cas9 technology[53]. Briefly, we targeted both the GFP-tag (to eliminate the GFP-GR transgene) and the endogenous GR by the non-homologous-end-joining method. Western blots show undetectable levels of the GR protein (Fig. 2a) and functional studies confirmed the lack of a Dex response (Fig. 2c). KOGR + HaloTag-GR cells were generated by transduction with a retrovirus carrying the HaloTag-GR transgene. Briefly, 5 million Phoenix A cells were plated in a 10 cm dish 24 h prior to transfection with 10 µg pRevTRE-HaloTag-GR using JetPRIME transfection reagent (Polyplus transfection) according to the manufacturer's recommended protocol. Virus containing supernatant was collected 48 h post transfection and filtered through a 0.45 µM filter. Filtered virus-containing Phoenix cell supernatant was diluted with an equal volume of fresh media and polybrene was added to a final concentration of 5 µg ml$^{-1}$. A volume of 2 ml of this virus solution was used to infect 200,000 KOGR cells. At 48 h post transduction, the cells were selected with 500 µg ml$^{-1}$ Hygromycin (Sigma-Aldrich).

**Plasmid constructs.** The pHaloTag-GR expresses the rat GR with HaloTag fused in the C-terminal domain under the CMVd1 promoter[10]. The pHalo-GRC440G and pHalo-GRA477T-I646A-C440G (GRmonC440G) were generated by using a QuikChange II XL Site-Directed Mutagenesis Kit according to the manufacturer's instructions (Stratagene, La Jolla, CA, USA). The pHaloTag-ER expresses the human ERα with HaloTag fused in the C-terminal domain under the CMVd1 promoter[16]. The pHaloTag-PR expresses the human PR isoform beta with HaloTag fused in the N-terminal domain under the CMV promoter. Construct (pFN21AB9766) was purchased from Promega (Madison, WI, USA). The pHaloTag-AR expresses the human AR with HaloTag fused in the C-terminal domain. Construct was custom-made by Promega. The pHaloTag-c-FOS expresses the rat c-FOS with HaloTag fused in the N-terminal domain under CMVd1 promoter. It has been generated by PCR amplification from pcDNA3-FLAG-Fos WT (Addgene plasmid #8966) and sub cloned into the pFN22K (Promega, Madison, WI, USA) backbone with SgfI and PmeI sites. The pEGFP-c-FOS expresses the rat c-FOS with EGFP fused in the N-terminal domain under the CMV promoter. It has been generated by PCR amplification from pcDNA3-FLAG-Fos WT and sub cloned into the pEGFP-C1 backbone (Clontech, Mountain View, CA, USA) with KpnI and BamHI sites. The pHaloTag-a-FOS expresses the human a-FOS with HaloTag fused in the N-terminal domain under CMVd1 promoter. It has been generated by PCR amplification from pRev-TRE-aFos and sub cloned into the pFN22K backbone with SgfI and PmeI sites. The pEGFP-a-FOS expresses the human a-FOS with EGFP fused in the N-terminal domain under the CMV promoter. It has been generated by PCR amplification from pRev-TRE-aFos and sub cloned into the pEGFP-C1 backbone with SalI and BamHI sites. The pHaloTag-c-JUN expresses the mouse c-JUN with HaloTag fused in the N-terminal domain under CMVd1 promoter. It has been generated by PCR amplification from Flag-JunWT-Myc (Addgene plasmid #47443) and sub cloned into the pFN22K backbone with SgfI and PmeI sites. The pHaloTag-GRIP1

expresses the mouse GRIP1 with HaloTag fused in the N-terminal domain under CMVd1 promoter. It has been generated by PCR amplification from pEGFP-GRIP1 and sub cloned into the pFN22K backbone with *SgfI* and *PmeI* sites. The pHaloTag-GRIP1mutant (GRIP1mut) expresses mouse GRIP1-L693A-L694A-L748A-L749A with HaloTag fused in the N-terminal domain under CMVd1 promoter. It has been generated by PCR amplification from pEGFP-GRIP1mut and sub cloned into the pFN22K backbone with *SgfI* and *PmeI* sites. The pSNAP-tag-BRG1 expresses the human BRG1 with SNAP-tag fused in the N-terminal domain under CMV promoter. It has been generated by PCR amplification of SNAP from pSNAPf (N9183S, New England Biolabs, Ipswich, MA, USA) and sub cloned into the mCherry-hsBRG1 backbone with *AgeI* and *SalI* sites. This sub-cloning replaced the mCherry sequence with the SNAP sequence. The pRevTRE-HaloTag-GR contains HaloTag-GR in tetracycline regulated retroviral vector under minimal CMV promoter. It has been generated by PCR amplification from pHaloTag-GR and sub cloned in to the pRevTRE (Clontech) backbone with *AgeI* and *MfeI* sites.

**Isolation of proteins and immunoblotting.** For protein isolation, cells were grown 24 h with or without tetracycline (for expression of integrated GFP-GR). Subsequently, the cells were washed with cold PBS and collected with PBS-containing protease inhibitors (EDTA-free complete protease inhibitor cocktail, Roche, Indianapolis, IN, USA). Cell pellets were suspended into RIPA buffer (50 mM Tris pH 8.0, 150 mM NaCl, 1% Tergitol, 0.5% Na-Deoxycholate, 0.1% SDS). Protein concentrations were measured by Bradford assay, and 30 μg of protein extracts were separated by electrophoresis on 4–20% Mini-PROTEAN TGX Stain-free gels (Bio-Rad, Hercules, CA, USA). Separated proteins were transferred to Trans-Blot Turbo PVDF membranes (Bio-Rad) using Trans-Blot Turbo transfer system (Bio-Rad) using manufacturer's instructions. After blocking with 5% milk in TBS + 0.3% Tween, the membranes were probed with the following primary antibodies: anti-GR (1:2,000, sc-1004, Santa Cruz Biotechnology, Santa Cruz, CA, USA), anti-GAPDH (1:7,500, ab8245, Abcam, Cambridge, MA, USA), and anti-GRIP1 (1:2,000, A300-346A, Bethyl Laboratories, Montgomery, TX, USA) in 5% milk in TBS + 0.3% Tween overnight with rocking at 4 °C. After four washes with TBS + 0.3% Tween, membranes were probed with HRP-conjugated secondary mouse or rabbit antibodies (1:2,500, 31,430 and 31,460, respectively; Pierce Thermo Scientific, Rockford, IL, USA) for 1 h in 5% milk in TBS + 0.3% Tween. After four washes with TBS + 0.3% Tween, the membranes were incubated with Super Signal Pico detection reagent (Pierce Thermo Scientific, Rockford, IL, USA) and visualized using ChemiDoc MP imaging system (Bio-Rad). Images of full membranes shown in Fig. 2a,b and Supplementary Fig. 4h can be found in Supplementary Figs 8 and 9, respectively.

**Isolation of RNA and RT–qPCR.** For RNA isolation, cells were grown for 24 h in DMEM medium containing 10% charcoal-stripped FBS. Cells were collected after 1 h of 100 nM Dex treatment. RNA was extracted using the PureLink RNA mini kit (Invitrogen) according to the manufacturer's instructions. Subsequently, cDNA was generated from 1 μg RNA using the iScript cDNA Synthesis Kit (Bio-Rad). Real-time quantitative PCR was performed using the iQ SYBR Green Supermix (Bio-Rad) on a CFX96 Touch Real-Time PCR Detection System (Bio-Rad). Primer sequences were designed to amplify only nascent RNA, using PCR amplicons that cross an exon–intron or untranslated region (UTR)-intron boundary. β-actin was used to normalized the data, which is expressed as fold induction relative to each cell line vehicle treatment. Primers used for quantitative PCR are: Per1, forward 5′-CTTCTGGCAATGGCAAGGACTC and reverse 5′-CAGCATCATGCCAT-CATACACACA-3′; Sgk, forward 5′-GAAACAGAGAAGGATGGGCCTGAAC-3′ and reverse 5′-GATCTCAGCTCCAGCACCACCAC-3′; Orm, forward 5′-ATTC-TTGTCATGGTGAGCCTCCTGC-3′ and reverse 5′-GCTCAGGGTCTCATTG-GTGATAGGG-3′; b-actin, forward 5′-GCTGGAAAAGAGCCTCAGGGC-3′ and reverse 5′-CGCATCCTCTTCCTCCCTGGAG-3′.

**RNA interference.** For depletion of GRIP1, 5 million cells were electroporated with 10 μg of ON-TARGETplus Non-Targeting Control Pool (siSCR, D-001810, Dharmacon, Lafayette, CO, USA) or ON-TARGETplus SMARTpool mouse Ncoa2 (siGRIP1, L-040667, Dharmacon) using BTX T820 Electro Square Porator (Harvard Apparatus, Holliston, MA, USA) with three 10 ms pulses using low voltage pulse length and 140 V peak pulse. The cells were grown for 48 h and subjected to a second round of electroporation as before. However, in the second electroporation, 1 μg of HaloTag-GR was also electroporated with additional siSCR or siGRIP1 to the cells. The cells were grown for 24 h, and subsequently either proteins or RNAs were isolated; or single-molecule tracking of GR was performed as indicated.

**Subcellular localization of FOS- and JUN-tagged proteins.** Images were taken at the CCR, LRBGE Optical Imaging Core facility (NIH, Bethesda, MD, USA) in an LSM 780 laser scanning microscope (Carl Zeiss, Inc., Thornwood, NY) equipped with an environmental chamber. We used a × 63 oil immersion objective (nmerical aperture = 1.4). The excitation source was 488 nm for GFP and 561 nm for JF$_{549}$. Fluorescence was detected with a GaAsP detector in photon-counting mode.

**Single-molecule tracking.** Cells were transiently transfected with the indicated pHaloTag-, pSNAP-tag- or GFP-fusion proteins using jetPRIME reagent (PolyPlus, New York, NY, USA), under conditions that do not generate overexpression of the HaloTag construct (Fig. 2a,b). Briefly, 500 ng of DNA was transfected to ~100,000 cells for 4 h. After an overnight recovery, 3617 cells were treated with 5 nM of the cell-permeable Janelia Fluor 549 (JF$_{549}$) HaloTag ligand[20] for 20 min. For GRKO + HaloTag-GR cells, where the expression level of the HaloTag-GR protein is similar to the endogenous GR (Fig. 2a), 0.5 nM of JF$_{549}$ was used because higher amounts of dye-generated excessive label density. For SNAP-tag, 3617 cells were treated with 250 nM of JF$_{549}$ SNAP-tag ligand for 40 min. All cells are then washed three times for 15 min with phenol red-free DMEM media (Invitrogen) to remove the unbound fluorescent molecules. Subsequently, the cells were treated with or without 100 nM of the indicated hormones for 20 min before imaging. In transient transfections, regardless of the DNA amount transfected, imaged cells had 30–100 diffraction-limited spots at the two-dimensional plane. This is a similar number of spots observed in cells with stably integrated HaloTag-GR treated with an order of magnitude less fluorescence dye than in transient transfections.

The custom-built microscope from the CCR, LRBGE Optical Microscopy Core facility is controlled by μManager software (Open Imaging, Inc., San Francisco, CA.), equipped with a × 150, 1.45 numerical aperture objective (Olympus Scientific Solutions, Waltham, MA), a 561 nm laser (iFLEX-Mustang, Excelitas Technologies Corp., Waltham, MA), an acousto-optic tunable filter (AOTFnC-400.650, AA Optoelectronic, Orsay, France) and HILO illumination[19]. Six-hundred frames of fluorescent images are collected at a rate of 5 Hz on an EM-CCD camera (Evolve 512, Photometrics), except the experiments on Fig. 3, where the collection rate (that is, interval time) was variable and set as indicated. In all cases, the exposure time was fixed and set to 10 ms. The particle tracking is performed with the 'TrackRecord' software developed in Matlab (The Matworks Inc.)[54]. The software combines a package for single-particle and single-molecule tracking[55] with the routines to isolate and analyse the behaviour of chromatin-bound molecules[56]. A region of interest encompassing the nuclear compartment is selected based on a maximum projection image from the 600 frames stack. Potential particles are located in each frame of the movie based on a user-defined intensity threshold after applying Wiener, top-hat and size filters that respectively remove speckle noise, correct for uneven illumination, and highlight features that are around 5 pixels in area. If multiple peaks are found within a radius of 7 pixels of each other, only the brightest pixel is kept. Each seed is then fit to a two-dimensional Gaussian to precisely determine its position. Particles in each frame are connected into trajectories using a nearest-neighbor algorithm[55] with molecules allowed to move a maximum of 4 pixels from 1 frame to the next, and only tracks that are at least 6 frames long are kept. Single-frame gaps in trajectories are filled with the average position of the particle in the existing flanking frames. The number of cells and tracks analysed per condition are described in Supplementary Data 1.

Residence times and bound fraction are determined by the fitting to the survival distribution[54]. Briefly, the survival histogram is generated from the track segments that each particle is stationary. In practice, even tightly bound particles move slightly due to chromatin and nuclear motion, and therefore a maximum frame-to-frame displacement of 220 nm, and a two-frame displacement of 270 nm ($r_{max}$) (both obtained from the motion of immobile histones) have been used to define bound portions of each particle's track. Because there is a chance that even a fast diffusing molecule will move less than these thresholds, a further constraint on the minimum number of time points in the bound segment for each particle ($N_{min}$) is used to reduce to <1% the contribution of diffusing molecules to the survival histogram[6]. The value used for $N_{min}$ depends on the frame rate, and those used for the various intervals are presented in Supplementary Data 1. The survival histogram is normalized to the total bound fraction, $B$, which is calculated by first isolating the localization events that have been assigned to bound molecules $N_B$ and by then dividing $N_B$ for the total number of detected particles (including those molecules not assigned to tracks because they appear for just a single frame). All survival histograms are corrected for photobleaching, which is characterized separately for each movie, by fitting the frame-dependent number of detected particles to a bi-exponential decay[10]. To extract residence times, the survival distribution, $S(t)$, is fit by least squares to a mixed exponential decay with two rate constants, $k_{ns} = 1/T_{ns}$ and $k_s = 1/T_s$:

$$S(t) = B \times (F_{ns} \exp(-k_{ns}t) + (1 - F_{ns}) \exp(-k_s t)),$$

where $B$ is the bound fraction, and $F_{ns}$ is the fraction of particles non-specifically bound. To check for over-fitting, the distribution is also fit to a single-component exponential:

$$S(t) = B \times \exp(-kt),$$

and to a three-component exponential, with two different specific binding decay constants, $k_{s1}$ and $k_{s2}$:

$$S(t) = B \times (F_{ns} \exp(-k_{ns}t) + F_{s1} \exp(-k_{s1}t) + (1 - F_{ns} - F_{s1}) \exp(-k_{s2}t)),$$

and the fits are compared using an *F*-test to ensure that the two-component model gives a significantly improved fit over the single- and three-component decay. The Box-plots represent the distribution of dwell times in the short-lived fast fraction ($F_{ns}$) and long-lived slow fraction ($1 - F_{ns}$) (Fig. 1b). Using the ratio acquired

from the exponential fitting between $F_{ns}$ and $1 - F_{ns}$ (Fig. 1c), the population of individual dwell times are divided into two populations, fast fraction (green boxes) and slow fraction (blue boxes). If the single-component exponential fit accurately explains the distribution, the individual dwell times are represented in one population (red boxes). Statistical comparisons between different residence times in the slow population are calculated using the Kolmogorov–Smirnov test (KS test). The entire distribution of dwell times is used in the statistical comparisons. Statistical comparisons between fast short-lived or slow long-lived fraction between conditions are calculated using the Student's $t$-test.

**Data availability.** The data sets generated during and/or analysed during the current study are available from the corresponding author upon reasonable request.

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

## Acknowledgements

This work was supported by the Intramural Research Program of the National Institutes of Health (NIH), the National Cancer Institute (NCI) and the Center for Cancer Research (CCR). V.P. was supported, in part, by the Sigrid Jusélius Foundation. We thank Jonathan B. Grimm and Luke D. Lavis for providing the Janelia Fluor dyes.

## Author contributions

V.P. and D.M.P. designed and performed all the experiments, and prepared the manuscript. G.L.H. initiated and directed the project. D.A.B. and T.S.K. provided imaging instrumentation, programming support and supported the data analysis. T.A.J. supported experimental design, execution and data analysis. R.L.S. and P.L. carried out cloning of the constructs. D.M. and T.M. constructed the HILO microscope.

## Additional information

**Competing interests:** The authors declare no competing financial interests.

