## [Peer Review File · Nature Communications]

Reviewers' comments:

Reviewer #1 (Remarks to the Author):

Paakinaho et al. show that transcription factors don't bind their chromatin target sequentially and stably for minutes or hours, but rather transiently for relatively short time durations in the range of seconds. This is demonstrated by state-of-the-art single molecule tracking experiments of the glucocorticoid receptor (GR) and several of its cofactors, and a sophisticated and careful data analysis. The work is focused on a controversial mechanism for the interaction of the GR with its target chromatin domain. The paper is certainly of great general interest to the community, since it addresses the fundamental question how gene activation is accomplished by transcription factors.

The employed single molecule tracking methods within living cells and the respective data analysis procedures are very sound and will represent a new standard in the field. Though the general idea of single molecule tracking of transcription factors in living cells has been realized before by the Hager lab and others, in this work a very careful and complete approach provides more and deeper insights into the dynamics of the GR and other transcription factors like the estrogen, progesterone and androgen receptors. According to the view of this reviewer the question of transcription factor dynamics in cell nuclei before and after their activation is resolved in this paper.

This work from the Hager lab is convincing, addresses a current and controversial subject and uses up-to-date techniques to gain novel conclusions. This reviewer clearly recommends a publication in Nature Communications.

Minor comments:

- (1) The authors do not report at all how they identified the nuclear interior as regions where they analyzed the TF dynamics.
- (2) For measurements with long interval times (Fig. 1a, Suppl. Fig 2) it is principally possible, that one receptor is replaced by another during the interval between acquisition of two images, which would lead to errors in the determined binding duration. The authors should comment on this.
- (3) Supplementary figure 5: the letter "e" for labeling the single panels is used twice (thus, the reference in line 254 is misleading)
- (4) Line 283, wording: "entertained"?

Reviewer #2 (Remarks to the Author):

This report extends extensive prior work from the Hager lab that established the rapid kinetics of dynamic interactions between the glucocorticoid receptor (GR) and target sites within the genome of live cells. Specifically the authors utilize state-of-the-art new technology for single molecule tracking (SMT) to uncover new mechanistic details of the dwell time of individual GR molecules on high affinity chromatin site and reveal for the first time distinctions between these dynamic parameters with the minor fraction of bulk GR in the nucleus. The author's analysis is quite detailed and the methodology described in excellent detail. While for most data rigorous statistical tests are employed to assess the significance of various dynamic parameters of chromatin association of GR and other factor there are a number of examples in the manuscript (indicated below) where the authors are too casual in their analysis of their results and do not apply appropriate statistical tests.

For example:

Line 128: "slight increase in residency time.." (Fig 2d and 2e)

Figure 3b: no statistical comparison between untreated long-lived fraction between different

steroid hormone receptors

Lines 269 & 270: "very slight changes ..in HaloGR residency times"..Is this "slight change" significant? In Figure 4d the authors show significance for a very modest ("slight"?) change in Dex effects on the long-live fraction of GRIP.

Additional concerns:

Lines 113-117: The data showing the presence of a long-lived fraction of the GR monomer mutant in the presence of Dex should be shown in the manuscript and not in a supplemental figure since it reveals a fundamental difference between the effect of Dex and Cort. Furthermore, the author's explanation is very unsatisfying. If the author's suggestion that long-lived binding of the monomeric GR upon Dex treatment is correct, does that imply that Cort bound monomeric GR does not have the capacity to tether? Has this been demonstrated in previous studies? If not, it shouldn't be too difficult given the expertise in the Hager lab to examine this in the cell line used in these studies.

Lines 190-192: (statement regarding unliganded action of AR and ER): This statement is oversimplified as the unliganded activity of various steroid hormone receptors may be cell-type specific and influenced by receptor expression levels, etc. Have the authors assessed the unliganded activity of AR and ER in the cell line used in these assays?

Lines 348-350: It is unclear why the authors chose this alternative explanation for their data (i.e. GRIP corepressor activity). Has GRIP been shown to function as a corepressor on GR target genes in the cell line used in this study? If not, this statement is potentially misleading and not a suitable alternative to explain the lack of an effect on GR transactivation upon GRIP ablation.

Reviewer #3 (Remarks to the Author):

Comments on "Single-molecule Analysis of Steroid Receptor and Cofactor Action in Living Cells" by Hager and co-workers.

This manuscript describes a comprehensive single-molecule in vivo study on the dynamic interaction between transcription factor and chromatin (and co-factors). Using HaloTag and time-lapse imaging, they measured the residence time of several hormone-dependent transcription factors under un-induced and induced conditions, and found that the interaction between TFs and DNA is highly dynamic and only a small fraction of TFs resides on the DNA at any moment. Interestingly, they found that the distribution of residence time depends on imaging parameters, cautioning the interpretation of data from different imaging conditions. Furthermore, using similar residence time quantification, the authors analyzed the interaction between glucocorticoid receptor and its cofactors and revealed a transient and dynamic nature of this interaction. This study is very thorough and quantifies the dynamic nature of interactions between TF and DNA as well as between TF and co-factors. The analysis is solid and the conclusions provide important insights into the dynamic nature of gene regulation, and should be of interest to the broad readership of this journal. Below are my comments:

1. Data quantification. The authors have been quantifying the data in terms of the residence time. However, they did not present another useful quantification of the dynamics, i.e., the intensity. The change in intensity could reveal more information other than photo bleaching, for example, whether the binding is cooperative and whether the intensity of the dots correlates with the residence time.

2. Potential artifacts in the assay. As I was reading the paper, I realized that some artifacts could have caused the change in residence time distribution, for example, hormone-induced TF

aggregation or ligand binding that reduces TF mobility.

3. A phenomenological model to explain distribution data from different imaging intervals would be very helpful. It seems that the authors already have enough data to construct a model that could potentially recover the "true" distribution of residence time instead of simple speculations. This model could transform our understanding of how imaging parameters influence data interpretation.

Reviewers' comments:

Reviewer #1 (Remarks to the Author):

Paakinaho et al. show that transcription factors don't bind their chromatin target sequentially and stably for minutes or hours, but rather transiently for relatively short time durations in the range of seconds. This is demonstrated by state-of-the-art single molecule tracking experiments of the glucocorticoid receptor (GR) and several of its cofactors, and a sophisticated and careful data analysis. The work is focused on a controversial mechanism for the interaction of the GR with its target chromatin domain. The paper is certainly of great general interest to the community, since it addresses the fundamental question how gene activation is accomplished by transcription factors. The employed single molecule tracking methods within living cells and the respective data analysis procedures are very sound and will represent a new standard in the field. Though the general idea of single molecule tracking of transcription factors in living cells has been realized before by the Hager lab and others, in this work a very careful and complete approach provides more and deeper insights into the dynamics of the GR and other transcription factors like the estrogen, progesterone and androgen receptors. According to the view of this reviewer the question of transcription factor dynamics in cell nuclei before and after their activation is resolved in this paper. This work from the Hager lab is convincing, addresses a current and controversial subject and uses up-to-date techniques to gain novel conclusions. This reviewer clearly recommends a publication in Nature Communications.

We thank the reviewer for the positive comments

Minor comments:

(1) The authors do not report at all how they identified the nuclear interior as regions where they analyzed the TF dynamics.

We use a maximum projection image from the 600 frames to visualize the nucleus and select a region of interest only within the nuclear compartment. We have now added this information in Methods (Page 18, Lines 494-496).

(2) For measurements with long interval times (Fig. 1a, Suppl. Fig 2) it is principally possible, that one receptor is replaced by another during the interval between acquisition of two images, which would lead to errors in the determined binding duration. The authors should comment on this.

We agree with the reviewer. The longer the interval the higher the probability that two "fast" molecules appear and disappear during the "dark times". A comment has been added to the main text (Page 6, Lines 155-157). In addition, due to the reviewer's interest in the different imaging intervals, we decided to move the data from the Supplements to a main Figure (Figure 3 in the revised manuscript).

(3) Supplementary figure 5: the letter "e" for labeling the single panels is used twice (thus, the reference in line 254 is misleading)

The histogram of HaloTag-cFOS was mislabeled. It should have been labeled as the letter “d” and the histogram for HaloTag-aFOS should have been labeled as the letter “e”. This has been corrected in the revised Supplementary Figure 4.

(4) Line 283, wording: “entertained”?
revised to “proposed”

Reviewer #2 (Remarks to the Author):

This report extends extensive prior work from the Hager lab that established the rapid kinetics of dynamic interactions between the glucocorticoid receptor (GR) and target sites within the genome of live cells. Specifically the authors utilize state-of-the-art new technology for single molecule tracking (SMT) to uncover new mechanistic details of the dwell time of individual GR molecules on high affinity chromatin site and reveal for the first time distinctions between these dynamic parameters with the minor fraction of bulk GR in the nucleus. The author’s analysis is quite detailed and the methodology described in excellent detail.

We thank the reviewer for his/her kind comments

While for most data rigorous statistical tests are employed to assess the significance of various dynamic parameters of chromatin association of GR and other factor there are a number of examples in the manuscript (indicated below) where the authors are too casual in their analysis of their results and do not apply appropriate statistical tests.

For example:

Line 128: “slight increase in residency time..” (Fig 2d and 2e)

We have now revised the text to indicate that there is a significant increase in residence time ($p < 0.001$, KS test) between transient and stable expression of HaloTag-GR (Page 5, Line 126). This is most likely due to different expression levels of HaloTag-GR between transient and stable conditions. Curiously, however, the difference in long-lived fraction is not significant ($p = 0.41$, t -test). This is also indicated in the revised text (Page 5, Line 127).

Figure 3b: no statistical comparison between untreated long-lived fraction between different steroid hormone receptors

We have now performed a statistical test (student’s t -test) to compare the untreated long-lived fraction between steroid receptors. Significant increase in long-lived fraction is observed with ER and AR ($p < 0.05$). This fits well with the observation that we stated in the original manuscript that both ER and AR showed a higher fraction of both short-lived and long-lived binding events in the untreated condition, compared to that of HaloTag alone (Supplementary Fig. 2a in revised manuscript). We have now indicated that the long-lived fraction of ER and AR is significantly higher than that of GR and PR (Page 7, Lines 189-190).

Lines 269 & 270: “very slight changes ..in HaloGR residency times” ..Is this “slight change” significant? In Figure 4d the authors show significance for a very modest (“slight”?) change in Dex effects on the long-live fraction of GRIP.

The change in residence time between HaloTag-GR+GFP vs. HaloTag-GR+GFP-cFOS is significant ($p=0.022$). This was shown in Fig. 6D, however, this was not written in the text. We have now added to the text that slight but significant changes were observed (Page 10, Lines 269-270). The statistical comparison of HaloTag-GR between siSCR and siGRIP1 is shown for long-lived fraction in Fig. 5g and for residence time in Fig. 5h in the revised manuscript. Both indicates that there is no significant difference.

Additional concerns:

Lines 113-117: The data showing the presence of a long-lived fraction of the GR monomer mutant in the presence of Dex should be shown in the manuscript and not in a supplemental figure since it reveals a fundamental difference between the effect of Dex and Cort.

As suggested by the reviewer, we have moved GRC440G and GRmonC440G in the presence of Dex to Figure 1 (Fig. 1d-f in the revised manuscript).

Furthermore, the author's explanation is very unsatisfying. If the author's suggestion that long-lived binding of the monomeric GR upon Dex treatment is correct, does that imply that Cort bound monomeric GR does not have the capacity to tether? Has this been demonstrated in previous studies? If not, it shouldn't be too difficult given the expertise in the Hager lab to examine this in the cell line used in these studies.

The reviewer has touched a very controversial topic in the GR field. In rigor, our data appears to indicate that a monomeric and DBD compromised Cort-GR complex does not bind DNA indirectly. This indeed suggests that Cort-GRmon is not able to tether, or at least is very inefficient at it. To the best of our knowledge, the capacity of a fully monomeric Cort-GR complex to transrepress (i.e. to tether) has not been tested. This is likely because transcriptional activity and GR oligomerization has no correlation as we previously demonstrated (Presman et al, Plos Biol 2014). Further, Cort-GR likely tethers naturally as a dimer (Figure 3 in Presman et al 2014) and not as a monomer as usually believed in the field. Functional examination on the prediction of these SMT results cannot be achieved only with the cell lines already used in this study. It is imperative to have a GR null background to functionally test tethering. Therefore, we would have to make stable cell lines from our GR KO cells with the different GR mutants and make a comprehensive functional characterization. Although we are currently developing these cell lines to further characterize genome-wide our oligomerization mutants, we feel this is beyond the scope of this manuscript and we will address it in a future publication.

Lines 190-192: (statement regarding unliganded action of AR and ER): This statement is oversimplified as the unliganded activity of various steroid hormone receptors may be cell-type specific and influenced by receptor expression levels, etc. Have the authors assessed the unliganded activity of AR and ER in the cell line used in these assays?

We agree with the reviewer that we may have overstated our conclusion since we did not assess the unliganded activity of AR or ER in our cell line. We have now rephrased our conclusion to "It is tempting to speculate this could reflect TF activity of the unliganded forms of ER and AR, as previously reported elsewhere" (Page 7, Lines 190-192)

Lines 348-350: It is unclear why the authors chose this alternative explanation for their data (i.e. GRIP corepressor activity). Has GRIP been shown to function as a corepressor on GR target genes in the cell line used in this study? If not, this statement is potentially misleading and not a suitable alternative to explain the lack of an effect on GR transactivation upon GRIP ablation.

We agree with the reviewer. We have eliminated the statement from the manuscript.

Reviewer #3 (Remarks to the Author):

Comments on “Single-molecule Analysis of Steroid Receptor and Cofactor Action in Living Cells” by Hager and co-workers.

This manuscript describes a comprehensive single-molecule in vivo study on the dynamic interaction between transcription factor and chromatin (and co-factors). Using HaloTag and time-lapse imaging, they measured the residence time of several hormone-dependent transcription factors under un-induced and induced conditions, and found that the interaction between TFs and DNA is highly dynamic and only a small fraction of TFs resides on the DNA at any moment. Interestingly, they found that the distribution of residence time depends on imaging parameters, cautioning the interpretation of data from different imaging conditions. Furthermore, using similar residence time quantification, the authors analyzed the interaction between glucocorticoid receptor and its cofactors and revealed a transient and dynamic nature of this interaction. This study is very thorough and quantifies the dynamic nature of interactions between TF and DNA as well as between TF and co-factors. The analysis is solid and the conclusions provide important insights into the dynamic nature of gene regulation, and should be of interest to the broad readership of this journal.

We thank the reviewer for the positive comments

Below are my comments:

1. Data quantification. The authors have been quantifying the data in terms of the residence time. However, they did not present another useful quantification of the dynamics, i.e., the intensity. The change in intensity could reveal more information other than photo bleaching, for example, whether the binding is cooperative and whether the intensity of the dots correlates with the residence time.

While we agree that using the intensity information could be very useful, unfortunately the HILO set-up does not allow us to do so. In HILO, the illumination profile is inclined, which means the laser does not hit the sample in a homogenous fashion. Therefore, the intensity of each diffraction-limited spot would depend on its position relative to the inclined sheet. We have discussed this in further detail in a recently accepted manuscript (Presman et al, Methods 2017) <http://dx.doi.org/10.1016/j.ymeth.2017.03.014>). Regarding changes in intensity within a single-spot, a hallmark of single molecules is that they present a step-wise photobleaching. We have used this property before (Presman et al, PNAS 2016) to infer the oligomerization state of the GR. However, this needs to be performed on fixed samples. Finally, proving cooperative binding is a current goal in

our lab. We are developing a two-color SMT assay to test cooperative binding of steroid receptors and cofactors.

2. Potential artifacts in the assay. As I was reading the paper, I realized that some artifacts could have caused the change in residence time distribution, for example, hormone-induced TF aggregation or ligand binding that reduces TF mobility.

We are not entirely sure what the reviewer means by “hormone-induced TF aggregation”.

If the reviewer means aggregation such as foci formation (van Steensel et al, Journal of Cell Science 1995) or polyQ-dependent aggregation as in the case of AR (Jochum et al, BBA 2012), then we do not believe it will constitute an artifact because even if they are visible at the lower labelling density used in our assays, then it would not behave as single-molecules. i.e. they will photobleach exponentially and not step-wise like. We have not seen any evidence of exponential photobleaching at individual diffraction-limited spots. However, we cannot exclude the possibility that the very long binding events (i.e. the outliers) correspond at least in part, to this group. Furthermore, it has been shown by single-molecule imaging that mutant Huntingtin protein forms visible aggregates differing from that of normal Huntingtin protein (Li et al, eLife 2016). Hence, aggregates should be visible in our assays.

If the reviewer is referring to aggregation as ligand induced oligomerization, then it is possible that higher oligomerization states will give an overestimation of the residence time. For example, a tetramer will have more chances of having at least one of the subunits surviving longer before it photobleaches than a monomeric protein. This could explain the differences observed between the stable HaloTag-GR cell line (no endogenous GR) vs the transient transfection data (presence of endogenous GR). Nevertheless, the low-density labelling conditions makes this argument very slim.

We also do not exactly understand the reviewer’s argument regarding TF mobility. It is true that ligand binding reduces GR mobility measured by FRAP (Schaaf et al, MCB 2003). However, interpretation of FRAP data has been debated for many years (Mueller et al, Curr Opin Cell Biol 2010) and even though there is no clear consensus, it seems that FRAP includes information about chromatin binding and mostly diffusing activity of a TF. Thus, why changes in TF mobility would create an artifact in SMT? It seems to us that ligand effects would not affect the determination of residence times but rather confirm previous results performed by FRAP (Mazza et al, NAR 2012).

3. A phenomenological model to explain distribution data from different imaging intervals would be very helpful. It seems that the authors already have enough data to construct a model that could potentially recover the “true” distribution of residence time instead of simple speculations. This model could transform our understanding of how imaging parameters influence data interpretation.

We agree that some type of model that can describe the different imaging interval data would greatly benefit the community, but our efforts in this direction have thus far not been productive. We have attempted to analyze the interval data in the manner of Normanno, et al Nat Comm 2015, who combined data taken at several imaging intervals to extract a single (non-specific) residence time. The presence of specific binding in our dataset complicates this analysis, and the results that

we obtained are not readily interpretable. Furthermore, when imaging at longer intervals (1-2.5 s), it is entirely possible that the signal from the same location in consecutive frames actually represents two different molecules, making comparisons even more difficult. Although we plan to continue looking at ways to integrate all the interval data to extract the true distribution, at this point we are not prepared to make any conclusions about that possibility. Due to the reviewer's interest in the different imaging intervals, we decided to move the data from the Supplements to actual Figure (Figure 3 in the revised manuscript).

REVIEWERS' COMMENTS:

Reviewer #1 (Remarks to the Author):

The authors responded to all my concerns to my satisfaction.

Reviewer #2 (Remarks to the Author):

The authors have addressed all of my concerns with the inclusion of new data or with modifications to the text.

Reviewer #3 (Remarks to the Author):

The authors have addressed my concerns and I recommend the publication of the manuscript.

Reviewers' comments:

Reviewer #1 (Remarks to the Author):

The authors responded to all my concerns to my satisfaction.

We thank the reviewer for helping us improving the manuscript.

Reviewer #2 (Remarks to the Author):

The authors have addressed all of my concerns with the inclusion of new data or with modifications to the text.

We thank the reviewer for helping us improving the manuscript.

Reviewer #3 (Remarks to the Author): The authors have addressed my concerns and I recommend the publication of the manuscript.

We thank the reviewer for helping us improving the manuscript.